

# Game theory interpretation of digital soil mapping convolutional neural networks

José Padarian, Alex B. McBratney, and Budiman Minasny

Sydney Institute of Agriculture & School of Life and Environmental Sciences, The University of Sydney, New South Wales, Australia

**Correspondence:** José Padarian (jose.padarian@sydney.edu.au)

**Abstract.** The use of complex models such as deep neural networks has yielded large improvements in predictive tasks in many fields including digital soil mapping. Once of the concerns about using these models is that they are perceived as black boxes with low interpretability. In this paper we introduce the use of game theory, specifically SHAP values, in order to interpret a digital soil mapping model. SHAP values represent the contribution of a covariate to the final model predictions.

We applied this method to a multi-task convolutional neural network trained to predict soil organic carbon of Chile. The results show the contribution of each covariate to the model predictions in three different contexts: a) at a local level, showing the contribution of the various covariates for a single prediction, b) a global understanding of the covariate contribution, and c) a spatial interpretation of their contributions. The latter constitutes a novel application of SHAP values and also the first detailed analysis of a model in a spatial context. The analysis of a SOC model in Chile corroborated that the model is capturing sensible

relationships between SOC and rainfall, temperature, elevation, slope and topographic wetness index. The results agree with commonly reported relationships, highlighting environmental thresholds that coincide with significant areas within the study area. This contribution addresses the limitations of the current interpretation of models in digital soil mapping, especially in a spatial context. We believe that SHAP values are a valuable tool that should be included within the DSM framework since they address the important concerns regarding the interpretability of more complex models. The model interpretation is a crucial

step that could lead to generating new knowledge to improve our understanding of soils.

## 1 Introduction

The use of statistical and machine learning (ML) methods to model the distribution of soil classes or properties with environmental factors is a core component of digital soil mapping (DSM). Since the processes driving pedogenesis are generally

complex, there is a general trend in DSM studies to increase the complexity of the models. More advanced models such as tree-like models, neural networks, etc., tend to deal better with the complex non-linearities present in the data, usually outperforming more traditional methods such as generalised linear models (Lamichhane et al., 2019; Padarian et al., 2020). By





increasing the complexity of the modelling framework, researchers are able reduce the need to over-simplify the target process. With that in mind, we introduced the use of a multi-task convolutional neural network (Padarian et al., 2019) which uses a window of pixels around a punctual soil observation as input instead of the single pixel intercepting its location, in order to better capture its spatial context. In addition, thanks to its multi-task design, the model is capable of predicting multiple

soil properties simultaneously, taking into account the interaction between them. The introduction of these two novel features yielded an error decrease of around 30% compared with a more conventional ML model (Cubist tree model).

Complex models such as deep convolutional neural networks (CNN) have been demonstrated to excel in predictive tasks in many fields of science (Anwar et al., 2018; Nash et al., 2018; Shen, 2018; Webb, 2018) but they have raised concerns about their interpretability and potential biases, especially since these type of algorithms are starting to be applied to assist decision-

making in domains with high human impact (Dressel and Farid, 2018). To tackle some of these concerns, in 2016, the European Union introduced a General Data Protection Regulation that includes policies that give users the "right to explanation" when an algorithmic decision that affects them has been made (Goodman and Flaxman, 2017). To address the interpretability issue, the ML community has responded by focussing part of its research on improving this aspect of ML models and many advances have been made more recently (Doshi-Velez and Kim, 2017).

In this paper, we explore one approach, namely the game theory, to interpret the CNN model for soil organic carbon (SOC) prediction in Chile reported in Padarian et al. (2019). Our main objective is to corroborate that the model is capturing sensible relationships between the target soil property and the covariates used to train the model. First, we give the rationale behind using game theory to explain a soil predictive model and we explain the main measure used to perform this task. Second, we introduce the CNN model and the data used to train it to then describe the different levels of interpretability that we explore, from local

(i.e. for a particular observation) to global explanations. Finally, we present and discuss the results obtained, exploring the effect of the different predictors, their interactions and the correspondence to known geographical patterns in the study area.

## 2 Game theory and SHAP values

From a game theory perspective, a modelling exercise may be rationalised as the superposition of multiple collaborative games where, in each game, agents (explanatory variables) strategically interact to achieve a goal — making a prediction for a single

observation. As a result of this collaboration, the agents receive a "payout" proportionate to their contribution. In this context, the total gain (or loss) resulting from the collaboration is the deviation of the prediction from the mean of the predictions for the complete dataset (expected value). Game theory is the mathematical study of such "games" and the interactions and strategies between the involved agents (Nash, 1950; Rasmusen, 1989).

One method to estimate the expected marginal contribution of a covariate among all possible covariates' combination is using

Shapley values (Shapley, 1953). To compute these values, it is required to fit a model $f_{S \cup \{i\}}$ including that covariate $i$ and another model $f_S$ withholding it. The difference between both models' predictions on the input $x$ is the marginal contribution of covariate $i$. When using more than one covariate, that contribution will depend on the interaction with the rest of the covariates, hence this procedure should be repeated for the complete power set of the covariates (all possible subsets $S \subseteq F$ including the





empty set and the set $F$ of all the covariates). The final covariate contribution $\phi_i \in \mathbb{R}$ is the weighted average of all marginal contributions:

$$\phi_i = \sum_{S \subseteq F \setminus \{i\}} \frac{|S|!(|F|-|S|-1)!}{|F|!} \left[ f_{S \cup \{i\}} \left( x_{S \cup \{i\}} \right) - f_S \left( x_S \right) \right]. \tag{1}$$

In this paper we use a modification of the original method proposed by Shapley (1953). Since the original method requires re-
training $2^{|F|}$ models, it quickly becomes prohibitive to apply on complex models and large datasets. By using techniques such as sampling approximations to Eq. 1 and approximating the effect of removing covariates by using other samples from the training dataset, Lundberg and Lee (2017) developed a more efficient method (Shapley additive explanations; SHAP) to estimate the contribution of covariates preserving the properties of Shapley values. The SHAP values have been used to describe models developed in many fields, including medicine (Lundberg et al., 2018), engineering (Parsa et al., 2020) and finances (Mokhtari
et al., 2019), showing a much stronger agreement with human explanations compared to alternative methods (Lundberg and Lee, 2017). Lundberg and Lee (2017) also adapted their method to be used on deep neural networks, which is what we used in this paper to interpret our CNN model. "Deep SHAP" solves the SHAP values of each component of the deep neural network by using linear approximations which are then aggregated by using a composition rule that enables the efficient approximation of SHAP values for the whole model.

The method is based on the concept that, in order to explain a complex model $f$, it is necessary to use an explanation model $g$, simpler than the original one. This simplification also includes the use of simplified inputs $x'$ that map to the original inputs $x$ through a mapping function $x = h_x(x')$, which is specific to $x$. Given a different set of inputs $z' \approx x'$, the method ensures that $g(z') \approx f(h_x(z'))$. In order to estimate the effect of varying the input $x$, $z'$ is obtained from reference values sampled from other observations from the training dataset. Then, the effect $\phi_i$ to each feature (Eq 1) becomes part of a linear function
of binary variables (i.e. explainable variables have two possible states — available or not):

$$g(z') = \phi_0 + \sum_{i=1}^{M} \phi_i z_i', \tag{2}$$

where $z' \in \{0,1\}^M$, and $M$ is the number of simplified inputs features. By summing all the features attribution $\phi_i z_i'$, including the attribution $\phi_0$ of the empty set (i.e. mean of the predictions for the complete dataset), it is possible to approximate the prediction $f(x)$ in what constitutes the additive feature attribution explanation model, which is on what many current
explanation methods are based Lundberg and Lee (2017).




## 3 Methods

### 3.1 Model and data description

The model that we explore in this paper corresponds to a multi-task CNN (Fig. 1), trained to simultaneously predict SOC content of Chile at five depth intervals (0–5, 5–15, 15–30, 30–60 and 60–100 cm). The input of the network are 3D arrays with shape $(29, 29, 5)$, which represent a 29x29 pixels window around each observation, for the five covariates used to explain the spatial distribution of SOC.

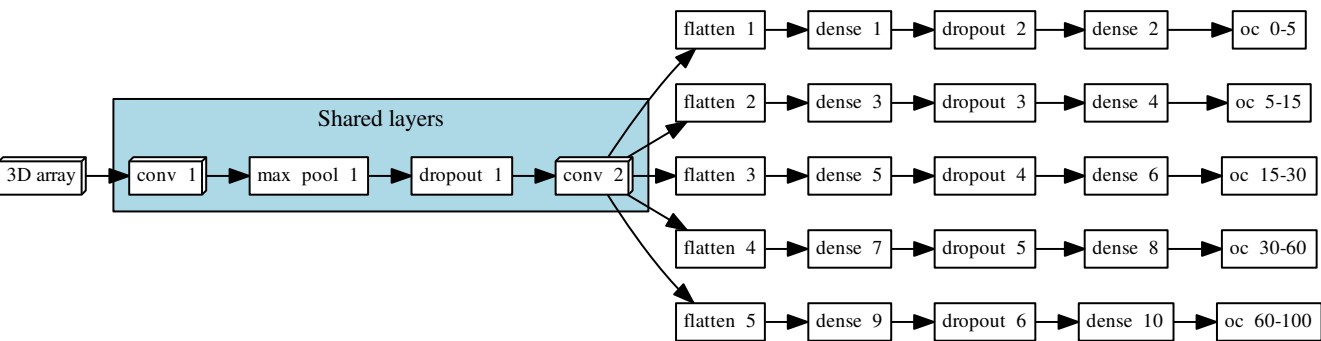

**Figure 1.** Architecture of the multi-task network. "Shared layers" represent the layers shared by all the depth ranges. Each branch, one per depth range, first flattens the information to a 1D array, followed by a series of 2 fully-connected layer and a fully-connected layer of size=1, which corresponds to the final prediction.

The national scale model was trained using a dataset of 485 soil profiles from Chile. As covariates, we used a) digital elevation model (HydroSHEDS, Lehner et al. (2008)), which are provided at 3 arc-second resolution, in addition to its derived slope and topographic wetness index, calculated using SAGA (Conrad et al., 2015); and b) long term mean annual temperature (MAT) and total annual rainfall (TAP) derived from information provided by WorldClim (Hijmans et al., 2005), at 30 arc-second resolution. All data layers were standardised to a 100 m grid size. For more details about the data and data preparation, including depth harmonisation and data augmentation, we refer the reader to Padarian et al. (2017) and Padarian et al. (2019).

### 3.2 Applying SHAP to a CNN soil model

To estimate the SHAP values, we used the *shap* Python library (https://github.com/slundberg/shap), using as background data (data to perform the approximations) the 436 samples corresponding to the CNN training dataset. In order to corroborate that the model is capturing sensible relationships between SOC and the different covariates, we illustrate the application of SHAP in four specific tasks, described bellow. All the figures correspond to the 0–5 cm depth range, unless stated otherwise.

**Single-sample (spatial):** Given the nature of the CNN input (a 29x29 context window), the first application of SHAP focussed on the contribution of each pixel to the final prediction. The obtained SHAP values should highlight influential areas in the landscape and the local influence of each covariate. It is important to remember that the covariates where resampled





to a 100 m grid size, hence the detailed, per-pixel contribution of TAP and MAT should be considered an artefact and it is probably better to only consider their aggregated value.

**Single-sample:** Since SHAP values have an additivity property (Lundberg and Lee, 2017), the per-pixel contributions can be aggregated into a single SHAP value per covariate. This aggregation provided a more intuitive measure of the contribution of the covariates since their values, once added to the expected value, should equal the predicted SOC content. We would like to remind the reader that SHAP values are not the amount of SOC that would vary if we remove a specific covariate since they are a weighted average of all marginal contributions (Eq. 1).

**Model:** Besides per-sample explanations, SHAP values can also be interpreted in a global context by analysing all the samples, simultaneously. This allowed us to explore the overall contribution of the different covariates and also the interactions between them. By having a global understanding of the relationships captured by the model, it was possible to analyse the model behaviour along environmental gradient and identify significant thresholds.

**Model (spatial):** To make a prediction at a single pixel, the CNN model uses a 29x29 pixels context window around the target pixel. In order to visualise the spatial pattern of the SHAP values, we estimated the SHAP values for the context windows around each pixel of a larger area. Since there is a considerable overlap between context windows, the final pixel contribution is an average of all the instances where that pixel is used as context (up to 841 times for a context window of size 29x29). Related to the previous task, this analysis allowed us to visualise the environmental thresholds in a spatial context. In order to compare the patterns captured by the CNN model, we also generated maps of the SHAP values obtained from a linear model and a tree-like model (Cubist). The Cubist model corresponds to the same model used to compare performance of the CNN reported in Padarian et al. (2019).

## 4 Results and discussion

If we only consider single samples that can be used as inputs to the model, it is possible to evaluate the contribution of each pixel to the model prediction. Fig 2 shows two samples extracted from two contrasting locations. The first sample (top panels) is close to 2,000 m.a.s.l. in a mountainous area, with a TAP of around 1,000 mm yr$^{-1}$ and MAT of 6 ℃. Consequently, it was possible to observe a negative contribution of the topography and a positive, large contribution of MAT. The second sample is located in a low valley, closer to the coast, with MAT of 12 ℃. In this case, the topography had a positive contribution (gentle slopes favouring deposition) and the effect of MAT is positive but lower than the first sample. In both cases, the MAT is lower compared with the rest of the country, which favours SOC accumulation (Krull et al., 2003).

In the case of the covariates related to topography, this analysis gives an indication of influential areas in the landscape (within the context window), either positive or negative, describing interactions over long distances in what Trenberth et al. (1998) denominated teleconnections. This somewhat contradicts the opinion of Behrens et al. (2019) where they state that it is not possible to extract teleconnections using CNNs.





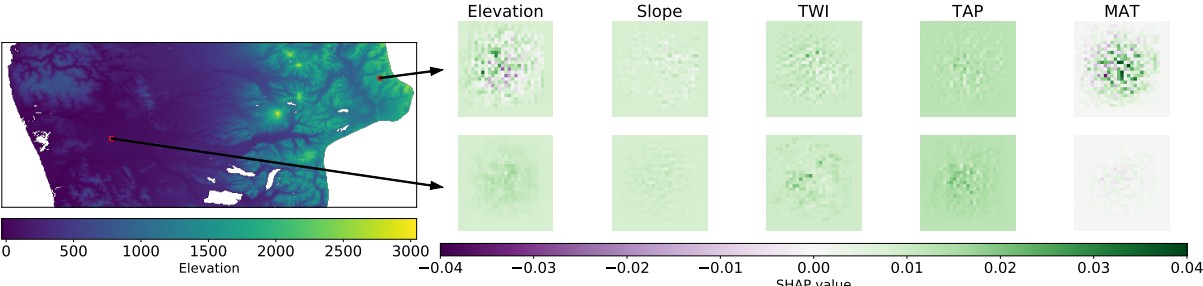

**Figure 2.** Spatial distribution of SHAP values for each covariate in two different samples (3D arrays used as inputs).

By aggregating the per-pixel contributions into a single SHAP value per sample (Fig 3), it is possible to corroborate the contributions observed in Fig 2, with the topography having a negative contribution in the first location (mountains), and an opposite effect in the second location (valley). Similar to MAT, the contribution of TAP in both cases is positive because this is a national model and both samples are located the southern part of the study area, which is more humid compared with the north.

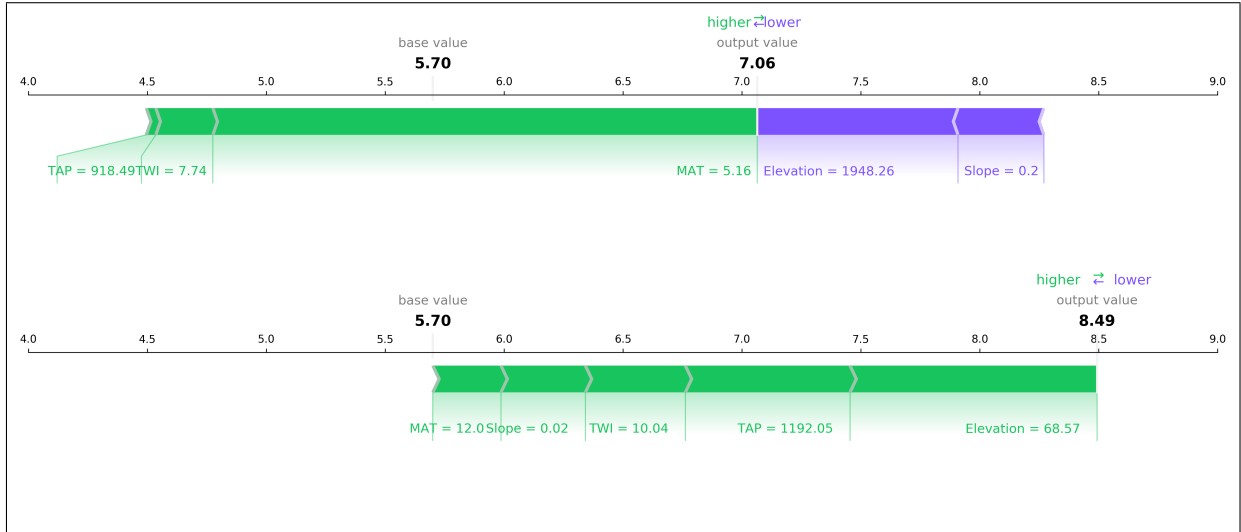

**Figure 3.** Force plots of the same two samples presented in Fig 2. The figures depict the positive (green) and negative (purple) aggregated contributions that deviate the SOC content from the expected base SOC value (the mean of the predictions for the complete dataset).

If we consider multiple samples simultaneously, it is possible to have a general idea of what the model has learned. We were able to confirm that the model captured the direct relationship between TAP and SOC and the inverse relationship between MAT and SOC (Fig. 4). Due to the great north-south climatic gradient in Chile and the abrupt changes in topography from the Andes to the coast, the main two drivers are TAP and MAP, with a broader range of SHAP values, closely followed by elevation. These results are expected and in line with those of previous studies at the national to continental scale (Martin et al.,

2011; Viscarra-Rossel et al., 2014; Akpa et al., 2016). In Fig. 4 it is also possible to corroborate that the predicted SOC content

decreased in depth.

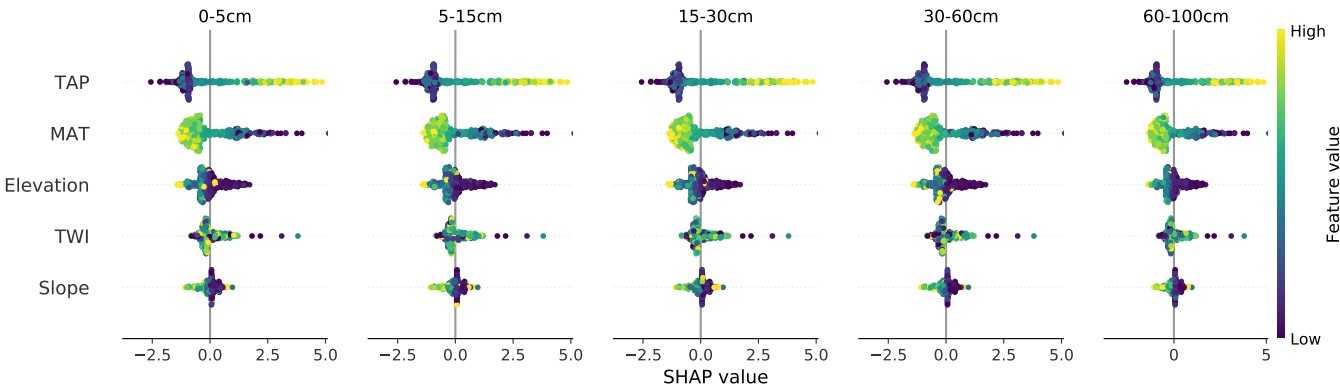

**Figure 4.** SHAP values for each covariate and soil depth interval for the CNN model. TAP: total annual precipitation; MAT: mean annual temperature; TWI: topographic wetness index.

It was also possible to further inspect the effect of particular covariates and evaluate if the model captured interactions between them. For instance, the positive contribution of temperature peaked in the range between 4 and 8 ºC with a negative

contribution above 12 ºC. Given the climatic conditions in the country, the decrease in MAT contribution is correlated with the decrease of precipitation, which is accentuated in areas with high temperatures (Fig. 5a) where carbon inputs are low (Ewing et al., 2008). The contribution of precipitation (Fig. 5a) showed a defined trend, with a mostly constant negative response up to around 1,000 mm yr$^{-1}$ where it starts increasing, becoming positive around 1,400 mm yr$^{-1}$. The model also captured an inverse relationship between SOC content and elevation (Fig. 5c) which higher values are usually associated with steeper

terrain, given the topography of the country. In general, this effect was exacerbated in areas with TAP higher than 400 mm yr$^{-1}$.

For both, temperature and precipitation, the threshold values where their contributions turn positive (around 12 ºC and 1,400 mm yr$^{-1}$, respectively) coincide with a significant area within the country (around 38º lat. S.), where Andisols become more prevalent and there is a change from a xeric to an udic soil moisture regime (Luzio, 2010), associated with a sharp increase in

the content of SOC (Padarian et al., 2017). In terms of the TAP threshold that seems to change the behaviour of the contribution of the elevation (around 400 mm yr$^{-1}$), it roughly corresponds to the transition between the arid and semi-arid zones of the country (Casanova et al., 2013), where the erosion processes by water start becoming more important.

All the previous interpretations are further corroborated by the resulting map of SHAP values (Fig. 6). It is possible to distinguish the valley close to the coast, which has lower TAP and slope, and higher MAT compared with the surrounding

areas. Those conditions contribute negatively to the model output, compared to the expected mean output. As mentioned before, even when those trends make sense in terms of SOC processes, they are probably interacting with the effect of human activity (agriculture) in the valley.

//doi.org/10.5194/soil-2020-17




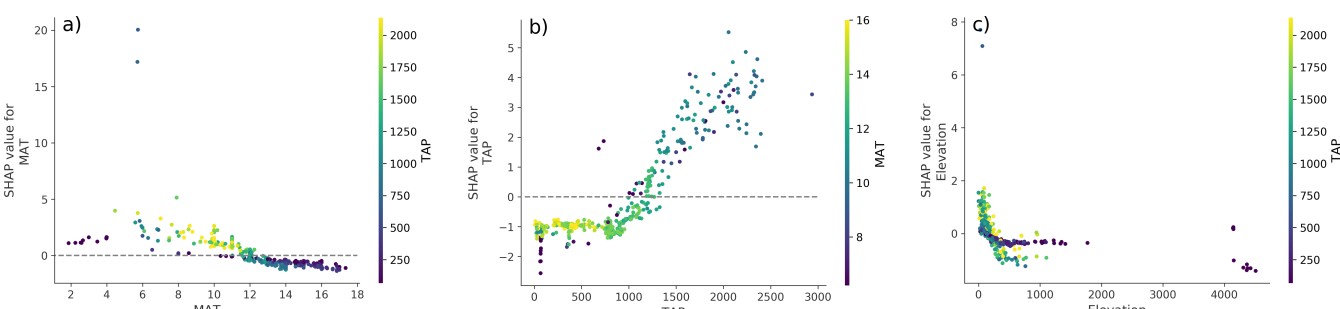

**Figure 5.** Dependency plots between SHAP values and selected covariates. a) Between SHAP values and mean annual temperature (MAT), showing the interaction with total annual precipitation (TAP; colour scale); b) between SHAP values and total annual precipitation , showing the interaction with mean annual temperature (colour scale); and c) Between SHAP values and elevation, showing the interaction with total annual precipitation (colour scale).

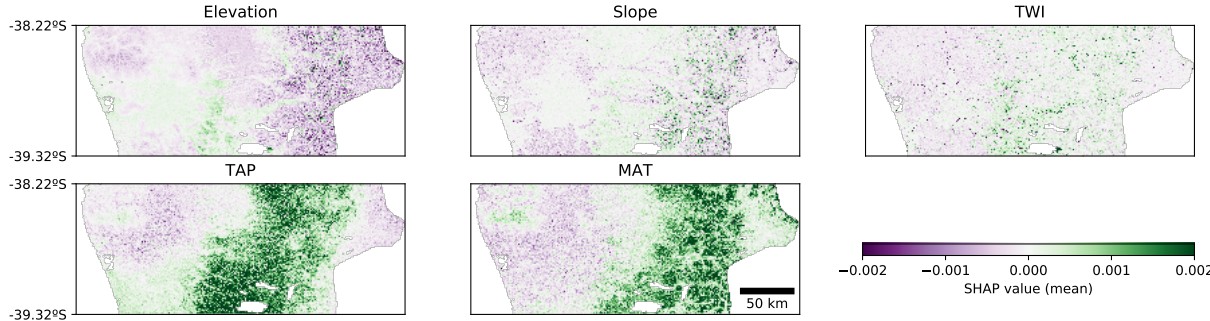

**Figure 6.** Spatial distribution of SHAP values for each covariate for the CNN model. The value of each pixel is an average of all the instances where that pixel is used as context (up to 841 times for a 29x29 context window). TAP: total annual precipitation; MAT: mean annual temperature; TWI: topographic wetness index.

The SHAP values can also be applied to other ML models and even to linear models. When comparing the SHAP values map of the CNN with the SHAP values map of a tree-like model and a linear model, it was possible to observe similar trends in all of them. In the tree-like model (Fig. 7) it is also possible to distinguish the valley close to the coast, the increase in the contribution of TAP and MAT, and the east-west gradient of the elevation contribution. The most evident difference is the
5    sharpness of the limits generated by the tree-like model, which is expected given the nature of the underlying algorithm. The linear model (Fig. 8) also showed similar spatial patterns but, as expected, much simpler and smoother than the tree-like and CNN models.

The results of this study show that it is possible to work towards interpretable deep learning models in DSM and that a complex model, generally perceived as a black box, can be inspected using SHAP values. It was possible to assess the covariates
10   importance for the whole model, providing an alternative to the variables of importance of the random forest algorithm, which is commonly used in DSM (Wiesmeier et al., 2011; Heung et al., 2014; Dharumarajan et al., 2017; Ellili et al., 2019). More




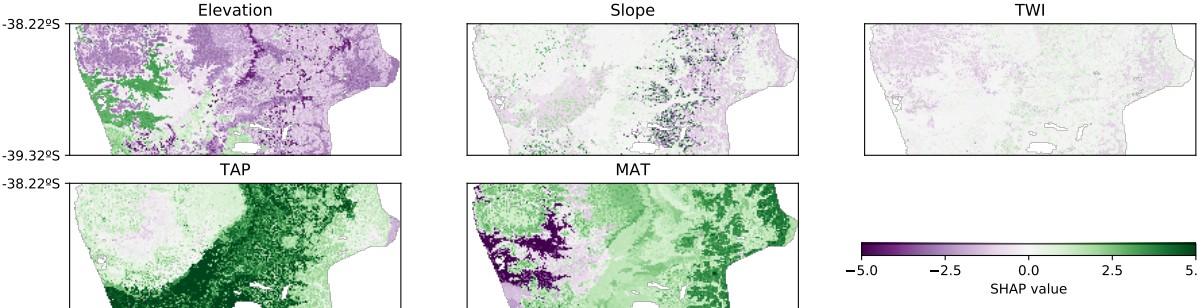

**Figure 7.** Spatial distribution of SHAP values for each covariate for the tree-like model. TAP: total annual precipitation; MAT: mean annual temperature; TWI: topographic wetness index.

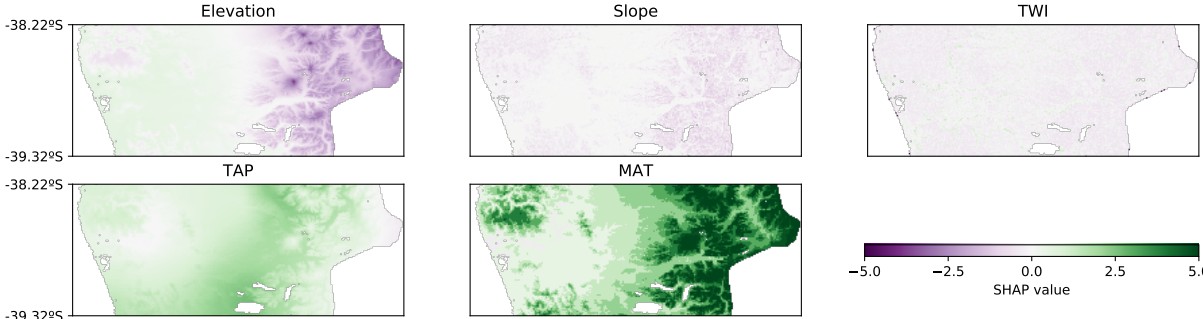

**Figure 8.** Spatial distribution of SHAP values for each covariate for the linear model. TAP: total annual precipitation; MAT: mean annual temperature; TWI: topographic wetness index.

importantly, we were able to evaluate the variable importance in a spatial context, generating maps of the contributions of each covariate to the model prediction, in a novel application of SHAP values. Surprisingly, considering the spatial nature of DSM, efforts to understand the behaviour of a model in space are scarce. To the best of our knowledge, this has only been attempted with tree-like models (Bui et al., 2006). These models generate rules to partition the data in different groups and,

5 when a partition is no longer possible, a linear regression is fitted to the observations of that terminal node. That allows the possibility of generating maps of a) the rules that lead to the terminal nodes, b) covariate thresholds defined in the partition rules, and c) the parameters of the terminal linear regressions. Given the linear nature of the tree-like models, the spatial representation of their rules, covariate thresholds, and parameters, although useful, usually leads to maps with abrupt changes or large continuous areas of dominance by a single rule. Also, since the model parameters are specific for each terminal node,

10 they are not comparable over the whole study area.

Using SHAP values shows promising results to interpret a DSM CNN, which it is not only necessary to corroborate that the model was trained properly, but it is also a fundamental part of the process leading to knowledge discovery (Fayyad et al., 1996). Here we presented a relatively simple model that only used five covariates but that already showed interesting




insights, identifying environmental thresholds that can be attributed to specific geographical areas. Potentially, including other covariates such as landuse and management practices could allow the model to extract knowledge related to how human intervention affects SOC content and perhaps could help us find solutions to prevent soil degradation. Of course, increasing the complexity of the problem generally translates into needing more data. In applications such as drug discovery, studies

reporting novel discoveries through deep learning methods use extensive databases (Zhang et al., 2017; Ekins et al., 2019; Zhavoronkov et al., 2019). On the other hand, global and national soil datasets are sparse in space and time, with a number of profiles in the order of hundreds to tens of thousands. We think that it is important to keep generating more soil data to avoid limiting the applicability of more complex deep learning models and their capacity to extract new insights that could improve our knowledge and understanding of soils.

## 10 5 Conclusions

In this paper we introduced the use of game theory, specifically SHAP values, in order to interpret a multi-task convolutional neural network trained to simultaneously predict soil organic carbon (SOC) content at five depth intervals. We illustrated how this method can be used to provide insights about the model. The results corroborated that the model captured sensible relationships between the target soil property and the covariates used to train the model.

We were able to interpret the contribution of the different covariates in three contexts. First, at a local level, showing the contribution of the covariates for a single prediction. Second, by analysing multiple local interpretations simultaneously, a global understanding of the covariates contribution. Third, a spatial interpretation of the contributions, which is a novel application of SHAP values and also the first detailed spatial application of this kind.

After a more detailed inspection of the contributions at the global level, we were able to identify environmental thresholds

consistent with significant areas within the study area. Those thresholds can be also inspected in a spatial context thanks to the map of SHAP values. This suggests that the modelling exercise, including data quality, model selection, and training, was successful.

Considering the limitations of the current interpretation of models in digital soil mapping, especially in a spatial context, we believe that SHAP values are a valuable tool that should be included in the DSM framework since they address the important

concerns regarding the interpretability of more complex models. Additionally, the insights provided from the ML models could also lead to knowledge discovery.

*Competing interests.* The authors declare that there are no conflicts of interest.

*Acknowledgements.* The authors acknowledge the University of Sydney HPC service at The University of Sydney for providing HPC resources that have contributed to the research results reported within this paper.



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
