# Peer review of "Game theory interpretation of digital soil mapping convolutional neural networks"

_SOIL, 2020_

## Referee Comment (RC1) · Anonymous Referee #1 · 24 Apr 2020

General comments

This paper introduces the use of the SHAP (Shapley additive explanations) coming from the Game Theory as a mean to assess the relative implications of covariates in a complex model from machine learning. The methodology is tested on a CNN (Convolutional Neural network) developed to model SOC content at 5 depth intervals in Chile.

The purpose of this study is significant giving the difficulty to interpret complex models developed on machine learning techniques, especially in a spatial context. The expression 'black boxes' describes those models pretty well. Here, the authors provide a solution to decrypt those black boxes.

[Figure]

The example provided here illustrates that SHAP could be promising allies in the future modelling works of many soil scientists, especially in a spatial context. Indeed, applying SHAP allowed the authors to confirm that the CNN model captured clear relationships (along with environmental thresholds) existing between SOC and different covariates, and this at the local and the global levels. SHAP also gave information of the relative implications of the covariates in the CNN model for different contexts.

This paper is of good scientific quality, well organized and written. The reading is fluid. I think this paper has its in SOIL after adding few complementary information.

Specific comments

It would be really appreciated having a bit more details in part 3.1, especially about the model functioning and the choice of the training and test sets.

In addition, it could be nice having few information about the validation of the model, its goodness-of-it, somewhere in the text before presenting the SHAP results, please. Maybe a simple point plot representing the predicted vs observed data based on the test set for at least the 0-5cm depth interval?

I just wonder why you did not directly input land cover/land use data in the model giving the important influence of this parameter on SOC content?

Technical corrections

p.2, l.1: 'researchers are able to reduce...'

p.3, l.25: The reference needs to be reformatted I think.

---

## Referee Comment (RC2) · Matt Aitkenhead (Referee) · 28 Apr 2020

[referee-annotated manuscript omitted]

---

## Author Comment (AC1) · 29 Apr 2020

**Response to reviewers**

Thanks both reviewers for their prompt responses. We will include your suggestions in the revised version of the manuscript.

**General comments**

Reading both reviews, the common thread is that more details are required in Section 3.1 (Model and data description). In this publication, we aim to introduce the use of SHAP to interpret a convolutional neural network (CNN) in the context of digital soil mapping. We used a model that was trained for a previous publication (Padarian et al., 2019) where we describe in detail all the rational, model training and validation. We agree with the reviewers and we will add more details to this manuscript, trying to strike a good balance between giving the reader relevant details (to this manuscript) and avoiding repetition. Note that both publications are open-access, hence the reader can reproduce the training and the interpretation without being limited by paywalls.

**Response to Anonymous Referee 1**

**I just wonder why you did not directly input land cover/land use data in the model giving the important influence of this parameter on SOC content?**

We agree on the great influence of land cover/use on SOC content but, sadly, the model was trained using legacy data which were collected during the 60s-90s and they do not have landcover information associated with them. Additionally, the profiles do not have a description/sampling date record. Chilean scientists are in the process of collating a more detailed soil database which will be eventually used to generate new national models which will be re-interpreted.

---

## Author Response (AR1)

**Response to reviewers**

Thanks both reviewers for their comments and corrections. We will include your suggestions in the revised version of the manuscript. Responses to general and specific comments below.

**General comments**

Reading both reviews, the common thread is that more details are required in Section 3.1 (Model and data description). In this publication, we aim to introduce the use of SHAP to interpret a convolutional neural network (CNN) in the context of digital soil mapping. We used a model that was trained for a previous publication (Padarian et al., 2019) where we describe in detail all the rational, model training and validation. We agree with the reviewers and we will add more details to the manuscript, trying to strike a good balance between giving the reader relevant details (to this manuscript) and avoiding repetition.

**Response to Anonymous Referee #1**

**I just wonder why you did not directly input land cover/land use data in the model giving the important influence of this parameter on SOC content?**

We agree on the great importance of land cover/use on SOC content but, sadly, the model was trained using legacy data which were collected during the 60s-90s and they do not have landcover information associated with them. Additionally, the profiles do not have a description/sampling date record. Chilean scientists are in the process of collating a more detailed soil database which will be eventually used to generate new national models which will be re-interpreted.

[revised manuscript text omitted]